# Decentralized Online Bandit Optimization on Directed Graphs with Regret Bounds

## Abstract

We consider a decentralized multiplayer game, played over $T$ rounds, with a leader-follower hierarchy described by a directed acyclic graph. For each round, the graph structure dictates the order of the players and how players observe the actions of one another. By the end of each round, all players receive a joint bandit-reward based on their joint action that is used to update the player strategies towards the goal of minimizing the joint pseudo-regret. We present a learning algorithm inspired by the single-player multi-armed bandit problem and show that it achieves sub-linear joint pseudo-regret in the number of rounds for both adversarial and stochastic bandit rewards. Furthermore, we quantify the cost incurred due to the decentralized nature of our problem compared to the centralized setting.

## 1 Introduction

Decentralized multi-agent online learning concerns agents that, simultaneously, learn to behave over time in order to achieve their goals. Compared to the single-agent setup, novel challenges are present as agents may not share the same objectives, the environment becomes non-stationary, and information asymmetry may exist between agents (Yang & Wang, 2020). Traditionally, the multi-agent problem has been addressed by either relying on a central controller to coordinate the agents' actions or to let the agents learn independently. However, access to a central controller may not be realistic and independent learning suffers from convergence issues (Zhang et al., 2019). To circumvent these issues, a common approach is to drop the central coordinator and allow information exchange between agents (Zhang et al., 2018; 2019; Cesa-Bianchi et al., 2021).

Decision-making that involves multiple agents is often modeled as a game and studied under the lens of game theory to describe the learning outcomes.[1] Herein, we consider games with a leader-follower structure in which players act consecutively. For two players, such games are known as Stackelberg games (Hicks, 1935). Stackelberg games have been used to model diverse learning situations such as airport security (Balcan et al., 2015), poaching (Sessa et al., 2020), tax planning (Zheng et al., 2020), and generative adversarial networks (Moghadam et al., 2021). In a Stackelberg game, one is typically concerned with finding the Stackelberg equilibrium, sometimes called Stackelberg-Nash equilibrium, in which the leader uses a mixed strategy and the follower is best-responding. A Stackelberg equilibrium may be obtained by solving a bi-level optimization problem if the reward functions are known (Schäfer et al., 2020; Aussel & Svensson, 2020) or, otherwise, it may be learnt via online learning techniques (Bai et al., 2021; Zhong et al., 2021), e.g., no-regret algorithms (Shalev-Shwartz, 2012; Deng et al., 2019; Goktas et al., 2022).

No-regret algorithms have emerged from the single-player multi-armed bandit problem as a means to alleviate the exploitation-exploration trade-off (Bubeck & Slivkins, 2012). An algorithm is called no-regret if the difference between the cumulative rewards of the learnt strategy and the single best action in hindsight is sublinear in the number of rounds (Shalev-Shwartz, 2012). In the multi-armed bandit problem, rewards may be adversarial (based on randomness and previous actions), oblivious adversarial (random), or stochastic (independent and identically distributed) over time (Auer et al., 2002). Different assumptions on the bandit rewards yield different algorithms and regret bounds. Indeed, algorithms tailored for one kind of rewards are sub-optimal for others, e.g., the Exp3 algorithm due to Auer et al. (2002) yields the optimal scaling for adversarial rewards but not for

---

[1]The convention is to use agents in learning applications and players in game theoretic applications, we shall use the game-theoretic nomenclature in the remainder of the paper.

stochastic rewards. For this reason, best-of-two-worlds algorithms, able to optimally handle both the stochastic and adversarial rewards, have recently been pursued and resulted in algorithms with close to optimal performance in both settings (Auer & Chiang, 2016; Wei & Luo, 2018; Zimmert & Seldin, 2021). Extensions to multiplayer multi-armed bandit problems have been proposed in which players attempt to maximize the sum of rewards by pulling an arm each, see, e.g., (Kalathil et al., 2014; Bubeck et al., 2021).

No-regret algorithms are a common element also when analyzing multiplayer games. For example, in continuous two-player Stackelberg games, the leader strategy, based on a no-regret algorithm, converges to the Stackelberg equilibrium if the follower is best-responding (Goktas et al., 2022). In contrast, if also the follower adopts a no-regret algorithm, the regret dynamics is not guaranteed to converge to a Stackelberg equilibrium point (Goktas et al., 2022, Ex. 3.2). In (Deng et al., 2019), it was shown for two-player Stackelberg games that a follower playing a, so-called, mean-based no-regret algorithm, enables the leader to achieve a reward strictly larger than the reward achieved at the Stackelberg equilibrium. This result does, however, not generalize to $n$-player games as demonstrated by D'Andrea (2022). Apart from studying the Stackelberg equilibrium, several papers have analyzed the regret. For example, Sessa et al. (2020) presented upper-bounds on the regret of a leader, employing a no-regret algorithm, playing against an adversarial follower with an unknown response function. Furthermore, Stackelberg games with states were introduced by Lauffer et al. (2022) along with an algorithm that was shown to achieve no-regret.

As the follower in a Stackelberg game observes the leader's action, there is information exchange. A generalization to multiple players has been studied in a series of papers (Cesa-Bianchi et al., 2016; 2020; 2021). In this line of work, players with a common action space form an arbitrary graph and are randomly activated in each round. Active players share information with their neighbors by broadcasting their observed loss, previously received neighbor losses, and their current strategy. The goal of the players is to minimize the network regret, defined with respect to the cumulative losses observed by active players over the rounds. The players, however, update their strategies according to their individually observed loss. Although we consider players connected on a graph, our work differs significantly from (Cesa-Bianchi et al., 2016; 2020; 2021), e.g., we allow only actions to be observed between players and player strategies are updated based on a common bandit reward.

**Contributions:** We introduce the *joint pseudo-regret*, defined with respect to the cumulative reward where all the players observe the same bandit-reward in each round. We provide an online learning-algorithm for general consecutive-play games that relies on no-regret algorithms developed for the single-player multi-armed bandit problem. The main novelty of our contribution resides in the joint analysis of players with coupled rewards where we derive upper bounds on the joint pseudo-regret and prove our algorithm to be no-regret in the stochastic and adversarial setting. Furthermore, we quantify the penalty incurred by our decentralized setting in relation to the centralized setting.

## 2 PROBLEM FORMULATION

In this section, we formalize the consecutive structure of the game and introduce the joint pseudo-regret that will be used as a performance metric throughout. We consider a decentralized setting where, in each round of the game, players pick actions consecutively. The consecutive nature of the game allows players to observe preceding players' actions and may be modeled by a DAG. For example, in Fig. 1, a seven-player game is illustrated in which player 1 initiates the game and her action is observed by players 2, 5, and 6. The observations available to the remaining players follow analogously. Note that for a two-player consecutive game, the DAG models a Stackelberg game.

We let $\mathcal{G} = (\mathcal{V}, \mathcal{E})$ denote a DAG where $\mathcal{V}$ denotes the vertices and $\mathcal{E}$ denotes the edges. For our setting, $\mathcal{V}$ constitutes the $n$ different players and $\mathcal{E} = \{(j, i) : j \to i, j \in \mathcal{V}, i \in \mathcal{V}\}$ describes the observation structure where $j \to i$ indicates that player $i$ observes the action of player $j$. Accordingly, a given player $i \in \mathcal{V}$ observes the actions of its direct parents, i.e., players $j \in \mathcal{E}_i = \{k : (k, i) \in \mathcal{E}\}$. Furthermore, each player $i \in \mathcal{V}$ is associated with a discrete action space $\mathcal{A}_i$ of size $A_i$. We denote by $\pi_i(t)$, the mixed strategy of player $i$ over the action space $\mathcal{A}_i$ in round $t \in [T]$ such that $\pi_i(t) = a$ with probability $p_{i,a}$ for $a \in \mathcal{A}_i$. In the special case when $p_{i,a} = 1$ for some $a \in \mathcal{A}_i$, the strategy is referred to as pure. Let $\mathcal{A}_\mathcal{B}$ denote the joint action space of players in a set $\mathcal{B}$ given by the Cartesian product $\mathcal{A}_\mathcal{B} = \prod_{i \in \mathcal{B}} \mathcal{A}_i$. If a player $i$ has no parents, i.e., $\mathcal{E}_i = \emptyset$, we use the convention $|\mathcal{A}_{\mathcal{E}_i}| = 1$.

We consider a collaborative setting with bandit rewards given by a mapping $r_t : \mathcal{A_V} \to [0,1]$ in each round $t \in [T]$. The bandit rewards may be either adversarial or stochastic. Let $\mathcal{C}$ denote a set of independent cliques in the DAG (Koller & Friedman, 2009, Ch. 2) and let $\mathcal{N}_k \in \mathcal{C}$ for $k \in [|\mathcal{C}|]$ denote the players in the $k$th clique in $\mathcal{C}$ with joint action space $\mathcal{A}_{\mathcal{N}_k}$ such that $\mathcal{N}_k \cap \mathcal{N}_j = \emptyset$ for $j \neq k$. For a joint action $\mathbf{a}(t) \in \mathcal{A_V}$, we consider bandit rewards given by a linear combination of the clique-rewards as

$$r_t(\mathbf{a}(t)) = \sum_{k=1}^{|\mathcal{C}|} \beta_k r_t^k(P^k(\mathbf{a}(t))), \tag{1}$$

where $r_t^k : \mathcal{A}_{\mathcal{N}_k} \to [0,1]$, $\beta_k \geq 0$ is the weight of the $k$th clique reward such that $\sum_{k=1}^{|\mathcal{C}|} \beta_k = 1$, and $P^k(\mathbf{a}(t))$ denotes the joint action of the players in $\mathcal{N}_k$. As an example, Fig. 1b highlights the cliques $\mathcal{C} = \{\{2,3,4\},\{1,5\},\{6\},\{7\}\}$ and we have, e.g., $\mathcal{N}_1 = \{2,3,4\}$, and $P^1(\mathbf{a}(t)) = (a_2(t), a_3(t), a_4(t))$. Note that each player influences only a single term in the reward (1).

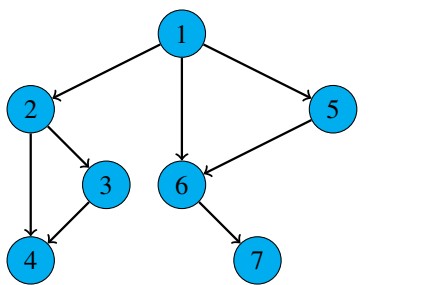

(a) A game with seven players.

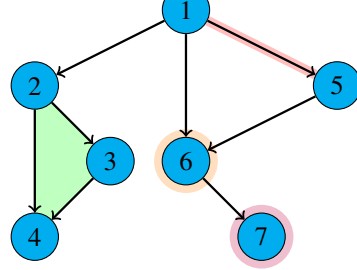

(b) Colored cliques comprising the bandit reward.

Figure 1: Game and reward structures.

In each round $t \in [T]$, the game proceeds as follows for player $i \in \mathcal{V}$: 1) the player is idle until the actions of all parents in $\mathcal{E}_i$ have been observed, 2) the player picks an action $a_i(t) \in \mathcal{A}_i$ according to its strategy $\pi_i(t)$, 3) once all the $n$ players in $\mathcal{V}$ have chosen an action, the player observes the bandit reward $r_t(\mathbf{a}(t))$ and updates its strategy. The goal of the game is to find policies $\{\pi_i(t)\}_{i=1}^n$ that depend on past actions and rewards in order to minimize the joint pseudo-regret $R(T)$ which is defined similarly to the pseudo regret (Shalev-Shwartz, 2012, Ch. 4.2) as

$$R(T) = r(\mathbf{a}^\star) - \mathbb{E}\left[\sum_{t=1}^T r_t(\mathbf{a}(t))\right], \quad r(\mathbf{a}^\star) = \max_{\mathbf{a} \in \mathcal{A_V}} \mathbb{E}\left[\sum_{t=1}^T r_t(\mathbf{a})\right], \tag{2}$$

where the expectations are taken with respect to the rewards and the player actions.[2] Note that $r(\mathbf{a}^\star)$ corresponds to the largest expected reward obtainable if all players use pure strategies. Hence, the pseudo-regret in (2) quantifies the difference between the expected reward accumulated by the learnt strategies and the reward-maximizing pure strategies in hindsight.

Our problem formulation pertains to a plethora of applications. Examples include resource allocation in cognitive radio networks where available frequencies are obtained via channel sensing (Janatian et al., 2015) and semi-autonomous vehicles with adaptive cruise control, i.e., vehicles ahead are observed before an action is decided (Marsden et al., 2001). Also recently, the importance of coupled rewards and partner awareness through implicit communications, e.g., by observation, has been highlighted in human-robot and human-AI collaborative settings (Bıyık et al., 2022).

As will be shown in the next section, any no-regret algorithm can be used as a building block for the games considered herein to guarantee a sub-linear pseudo-regret in the number of rounds $T$. As our goal is to study the joint pseudo-regret (2) for both adversarial and stochastic rewards, we resort to best-of-two-worlds algorithms for the multi-armed bandit problem (Bubeck & Slivkins, 2012). In particular, we will utilize the TSALLIS-INF algorithm that guarantees a close-to-optimal pseudo-regret for both stochastic and adversarial bandit-rewards (Zimmert & Seldin, 2021). Note that our analysis in the next section pertains to adversarial rewards; for stochastic rewards, our analysis apply verbatim by replacing Theorem 1 by (Zimmert & Seldin, 2021, Th. 4).

---

[2]This is called pseudo-regret as $r(\mathbf{a}^\star)$ is obtained by a maximization *outside* of the expectation.

## 3 ANALYSIS OF THE JOINT PSEUDO-REGRET

Our analysis of the joint pseudo-regret builds upon learning algorithms for the single-player multi-armed bandit problem. First, let us build intuition on how to use a multi-armed bandit algorithm in the DAG-based game described in Section 2. Consider a 2-player Stackelberg game where the players choose actions from $\mathcal{A}_1$ and $\mathcal{A}_2$, respectively, and where player 2 observes the actions of player 1. For simplicity, we let player 1 use a mixed strategy whereas player 2 is limited to a pure strategy. Furthermore, consider the rewards to be *a priori* known by the players and let $T = 1$ for which the Stackelberg game may be viewed as a bi-level optimization problem (Aussel & Svensson, 2020). In this setting, the action of player 1 imposes a Nash game on player 2 whom attempts to play optimally given the observation. Hence, player 2 has $A_1$ pure strategies, one for each of the $A_1$ actions of player 1.

We may generalize this idea to the DAG-based multiplayer game with unknown bandit-rewards and $T \geq 1$ to achieve no-regret. Indeed, a player $i \in \mathcal{V}$ may run $|\mathcal{A}_{\mathcal{E}_i}|$ different multi-armed bandit algorithms, one for each of the joint actions of its parents. Algorithm 1 illustrates this idea in conjunction with the TSALLIS-INF update rule introduced by Zimmert & Seldin (2021), which is given in Algorithm 2 for completeness.[3] In particular, for the 2-player Stackelberg game, the leader runs a single multi-armed bandit algorithm whereas the follower runs $A_1$ learning algorithms. For simplicity, Algorithm 1 assumes that player $i$ knows the size of the joint action space of its parents, i.e., $|\mathcal{A}_{\mathcal{E}_i}|$. Dropping this assumption is straightforward: simply keep track of the observed joint actions and initiate a new multi-armed bandit learner upon a unique observation.

---

**Algorithm 1** Learning algorithm of player $i \in \mathcal{V}$

---

Note: for ease of notation, let the actions in $\mathcal{A}_{\mathcal{E}_i}$ be labeled as $1, 2, \ldots, |\mathcal{A}_{\mathcal{E}_i}|$
1: **procedure**
2:     initialize cumulative rewards $\mathbf{L}_j \leftarrow \mathbf{0} \in \mathbb{R}^{A_i}$ for all $j \in [|\mathcal{A}_{\mathcal{E}_i}|]$
3:     initialize fixed-point $x_j \leftarrow 0$ for all $j \in [|\mathcal{A}_{\mathcal{E}_i}|]$
4:     initialize counter $n_j \leftarrow 0$ for all $j \in [|\mathcal{A}_{\mathcal{E}_i}|]$
5:     **for** $t = 1, 2, \ldots, T$ **do**
6:         observe the joint action $j \in [|\mathcal{A}_{\mathcal{E}_i}|]$ of the preceding players
7:         increase counter $n_j \leftarrow n_j + 1$
8:         obtain new strategy and new fixed-point $(\pi_i(t), x_j) \leftarrow$ TSALLIS-INF$(n_j, \mathbf{L}_j, x_j)$
9:         play action $a_i(t) \sim \pi_i(t)$
10:        observe the joint bandit-reward $r_t(\mathbf{a}(t))$
11:        update the cumulative loss $L_{j,k} \leftarrow L_{j,k} + \mathbf{1}\{a_i(t) = k\}(1 - r_t(\mathbf{a}(t)))/p_k$ for all $k \in [A_i]$
12:    **end for**
13: **end procedure**

---

**Algorithm 2** Strategy update for player $i \in \mathcal{V}$

---

**Input** time step $t$, cumulative rewards $\mathbf{L} \in \mathbb{R}_+^{A_i}$, previous fixed point $x$
**Output** strategy $\pi_i(t)$, fixed point $x$
1: **procedure** TSALLIS-INF
2:     set learning rate $\eta \leftarrow 2\sqrt{1/t}$
3:     **repeat**
4:         $p_j \leftarrow 4(\eta(L_j - x))^{-2}$ for all $j \in [A_i]$
5:         $x \leftarrow x - \left(\sum_{j=1}^{A_i} p_j - 1\right) / \left(\eta \sum_{j=1}^{A_i} p_j^{3/2}\right)$
6:     **until** convergence
7:     update strategy $\pi_i(t) \leftarrow (p_1, \ldots, p_{A_i})$
8: **end procedure**

---

Next, we go on to analyze the joint pseudo-regret of Algorithm 1. First, we present a result on the pseudo-regret for the single-player multi-armed bandit problem that will be used throughout.

---

[3] The original TSALLIS-INF Algorithm is given in terms of losses. To use rewards, one may simply use the relationship $l = 1 - r$.

**Theorem 1** (Pseudo-regret of TSALLIS-INF). *Consider a single-player multi-armed bandit problem with $A_1$ arms, played over $T$ rounds. Let the player operate according to Algorithm 1. Then, the pseudo-regret satisfies*

$$R(T) \leq 4\sqrt{A_1 T} + 1.$$

*Proof.* For a single player, $\mathcal{E}_1 = \emptyset$ and we have $|\mathcal{A}_{\mathcal{E}_1}| = 1$ by convention. Hence, our setting becomes equivalent to that of Zimmert & Seldin (2021, Th 1) and the result follows thereof. $\square$

Next, we consider a two-player Stackelberg game with joint bandit-rewards defined over a two-player clique. We have the following upper bound on the joint pseudo-regret.

**Theorem 2** (Joint pseudo-regret over cliques of size 2). *Consider a 2-player Stackelberg game with bandit-rewards, given by (1), defined over a single clique containing both players. Furthermore, let each of the players follow Algorithm 1. Then, the joint pseudo-regret satisfies*

$$R(T) \leq 4\sqrt{A_1 A_2 T} + 4\sqrt{A_1 T} + A_1 + 1.$$

*Proof.* Without loss of generality, let player 2 observe the actions of player 1. Let $a_1(t) \in \mathcal{A}_1$ and $a_2(t) \in \mathcal{A}_2$ denote the actions of player 1 and player 2, respectively, at time $t \in [T]$ and let $a_1^\star$ and $a_2^\star(a_1)$ denote the reward-maximizing pure strategies of the players in hindsight, i.e.,

$$a_1^\star = \arg\max_{a_1 \in \mathcal{A}_1} \mathbb{E}\left[\sum_{t=1}^{T} r_t(a_1, a_2^\star(a_1))\right], \quad a_2^\star(a_1) = \arg\max_{a_2 \in \mathcal{A}_2} \mathbb{E}\left[\sum_{t=1}^{T} r_t(a_1, a_2)\right]. \quad (3)$$

Note that the optimal joint decision in hindsight is given by $(a_1^\star, a_2^\star(a_1^\star))$. The joint pseudo-regret is given by

$$R(T) = \sum_{t=1}^{T} \mathbb{E}\left[r_t(a_1^\star, a_2^\star(a_1^\star)) - r_t(a_1^\star, a_2(t)) + r_t(a_1^\star, a_2(t)) - r_t(a_1(t), a_2(t))\right]$$

$$\leq \sum_{t=1}^{T} \max_{a_t \in \mathcal{A}_1} \mathbb{E}\left[r_t(a_t, a_2^\star(a_t)) - r_t(a_t, a_2(t))\right] + \mathbb{E}\left[\sum_{t=1}^{T} r_t(a_1^\star, a_2(t)) - r_t(a_1(t), a_2(t))\right]. \quad (4)$$

Next, let

$$a_1^+(t) = \arg\max_{a_t \in \mathcal{A}_1} \mathbb{E}\left[r_t(a_t, a_2^\star(a_t)) - r_t(a_t, a_2(t))\right]$$

and let $\mathcal{T}_a = \{t : a_1^+(t) = a\}$, for $a \in \mathcal{A}_1$, denote all the rounds that player 1 chose action $a$ and introduce $T_a = |\mathcal{T}_a|$. Then, the first term in (4) is upper-bounded as

$$\sum_{t=1}^{T} \max_{a_t \in \mathcal{A}_1} \mathbb{E}\left[r_t(a_t, a_2^\star(a_t)) - r_t(a_t, a_2(t))\right] = \sum_{a \in A_1} \sum_{t \in \mathcal{T}_a} \mathbb{E}\left[r_t(a, a_2^\star(a)) - r_t(a, a_2(t))\right]$$

$$\leq \sum_{a \in A_1} 4\sqrt{A_2 T_a} + 1 \quad (5)$$

$$\leq \max_{\sum_a T_a = T} \sum_{a \in A_1} 4\sqrt{A_2 T_a} + 1$$

$$= 4\sqrt{A_1 A_2 T} + A_1 \quad (6)$$

where (5) follows from Theorem 1 and because player 2 follows Algorithm 1.

Next, we consider the second term in (4). Note that, according to (3), $a_1^\star$ is obtained from the optimal pure strategies in hindsight of both the players. Let

$$a_1^\circ = \arg\max_{a_1 \in \mathcal{A}_1} \sum_{t=1}^{T} \mathbb{E}\left[r_t(a_1, a_2(t))\right]$$

and observe that $\mathbb{E}\left[\sum_{t=1}^{T} r_t(a_1^\star, a_2(t))\right] \leq \mathbb{E}\left[\sum_{t=1}^{T} r_t(a_1^\circ, a_2(t))\right]$. Hence, by adding and subtracting $r_t(a_1^\circ, a_2(t))$ to the second term in (4), we get

$$\mathbb{E}\left[\sum_{t=1}^{T} r_t(a_1^\star, a_2(t)) - r_t(a_1(t), a_2(t))\right] \leq \mathbb{E}\left[\sum_{t=1}^{T} r_t(a_1^\circ, a_2(t)) - r_t(a_1(t), a_2(t))\right]$$
$$\leq 4\sqrt{A_1 T} + 1 \tag{7}$$

where the last equality follows from Theorem 1. The result follows from (6) and (7). □

From Theorem 2, we note that the joint pseudo-regret scales with the size of the joint action space as $R(T) = \mathcal{O}(\sqrt{A_1 A_2 T})$. This is expected as a centralized version of the cooperative Stackelberg game may be viewed as a single-player multi-armed bandit problem with $A_1 A_2$ arms where, according to Theorem 1, the pseudo-regret is upper-bounded by $4\sqrt{A_1 A_2 T} + 1$. Hence, from Theorem 2, we observe a penalty of $4\sqrt{A_1 T} + A_1$ due to the decentralized nature of our setup. Moreover, in the centralized setting, Algorithm 2 was shown in Zimmert & Seldin (2021) to achieve the same scaling as the lower bound in Cesa-Bianchi & Lugosi (2006, Th. 6.1). Hence, Algorithm 1 achieves the optimal scaling. Next, we extend Theorem 2 cliques of size larger than two.

**Theorem 3** (Joint pseudo-regret over a clique of arbitrary size). *Consider a DAG-based game with bandit rewards given by (1), defined over a single clique containing $m$ players. Let each of the players operate according to Algorithm 1. Then, the joint pseudo-regret satisfies*

$$R(T) \leq 4\sqrt{T} \sum_{i=1}^{m} \prod_{k=1}^{i} \sqrt{A_k} + \sum_{i=1}^{m-1} \prod_{k=1}^{i} A_k + 1.$$

*Proof.* Let $R_{\mathrm{ub}}(T, m)$ denote an upper bound on the joint pseudo-regret when the bandit-reward is defined over a clique containing $m$ players. From Theorem 1 and Theorem 2, we have that

$$R_{\mathrm{ub}}(T, 1) = 4\sqrt{A_1 T} + 1$$
$$R_{\mathrm{ub}}(T, 2) = 4\sqrt{A_1 T} + 4\sqrt{A_1 A_2 T} + A_1 + 1,$$

respectively. Therefore, we form an induction hypothesis as

$$R_{\mathrm{ub}}(T, m) = 4\sqrt{T} \sum_{i=1}^{m} \prod_{k=1}^{i} \sqrt{A_k} + \sum_{i=1}^{m-1} \prod_{k=1}^{i} A_k + 1. \tag{8}$$

Assume that (8) is true for a clique containing $m - 1$ players and add an additional player, assigned player index 1, whose actions are observable to the original $m - 1$ players. The $m$ players now form a clique $\mathcal{C}$ of size $m$. Let $\mathbf{a}(t) \in \mathcal{A}_{\mathcal{C}}$ denote the joint action of all the players in the clique at time $t \in [T]$ and let $\mathbf{a}_{-i}(t) = (a_1(t), \ldots, a_{i-1}(t), a_{i+1}(t), \ldots, a_m(t)) \in \mathcal{A}_{\mathcal{C} \setminus i}$ denote the joint action excluding the action of player $i$. Furthermore, let

$$a_1^\star = \arg \max_{a_1 \in \mathcal{A}_1} \mathbb{E}\left[\sum_{t=1}^{T} r_t(a_1, \mathbf{a}_{-1}^\star(a_1))\right]$$

$$\mathbf{a}_{-1}^\star(a_1) = \arg \max_{\mathbf{a} \in \mathcal{A}_{\mathcal{C} \setminus 1}} \mathbb{E}\left[\sum_{t=1}^{T} r_t(a_1, \mathbf{a})\right]$$

denote the optimal actions in hindsight of player 1 and the optimal joint action of the original $m - 1$ players given the action of player 1, respectively. The optimal joint action in hindsight is given as

$\mathbf{a}^\star = (a_1^\star, \mathbf{a}_{-1}^\star(a_1^\star))$. Following the steps in the proof of Theorem 2 verbatim, we obtain

$$R(T) = \sum_{t=1}^{T} \mathbb{E}\left[r_t(\mathbf{a}^\star) - r_t(a_1^\star, \mathbf{a}_{-1}(t)) + r_t(a_1^\star, \mathbf{a}_{-1}(t)) - r_t(\mathbf{a}(t))\right]$$

$$\leq \sum_{t=1}^{T} \max_{a_1} \mathbb{E}\left[r_t(a_1, \mathbf{a}_{-1}^\star(a_1)) - r_t(a_1, \mathbf{a}_{-1}(t))\right] + \sum_{t=1}^{T} \mathbb{E}\left[r_t(a_1^\star, \mathbf{a}_{-1}(t)) - r_t(a_1(t), \mathbf{a}_{-1}(t))\right]$$

$$\leq \sum_{a \in \mathcal{A}_1} \sum_{t \in \mathcal{T}_a} \mathbb{E}\left[r_t(a, \mathbf{a}_{-1}^\star(a)) - r_t(a, \mathbf{a}_{-1}(t))\right] + \sum_{t=1}^{T} \mathbb{E}\left[r_t(a_1^\circ, \mathbf{a}_{-1}(t)) - r_t(a_1(t), \mathbf{a}_{-1}(t))\right]$$

$$\leq \sum_{a \in \mathcal{A}_1} R_{\mathrm{ub}}(T_a, m-1) + 4\sqrt{A_1 T} + 1$$

$$\leq A_1 R_{\mathrm{ub}}(T/A_1, m-1) + 4\sqrt{A_1 T} + 1 \tag{9}$$

where $\mathcal{T}_a$, $T_a$, and $a_n^\circ$ are defined analogously as in the proof of Theorem 2. By using the induction hypothesis (8) in (9) and by accounting for the original $m-1$ players being indexed from 2 to $m$, we obtain

$$R(T) \leq A_1 \left(4\sqrt{T/A_1} \sum_{i=2}^{m} \prod_{k=2}^{i} \sqrt{A_k} + \sum_{i=2}^{m-1} \prod_{k=2}^{i} A_k + 1\right) + 4\sqrt{A_1 T} + 1 = R_{\mathrm{ub}}(T, m)$$

which is what we wanted to show. $\qquad\square$

As in the two-player game, the joint pseudo-regret of Algorithm 1 achieves the optimal scaling, i.e., $R(T) = \mathcal{O}(\sqrt{T} \prod_{k=1}^{m} \sqrt{A_k})$, but exhibits a penalty due to the decentralized setting which is equal to $4\sqrt{T} \sum_{i=1}^{m-1} \prod_{k=1}^{i} \sqrt{A_k} + \sum_{i=1}^{m-2} \prod_{k=1}^{i} A_k$.

Up until this point, we have considered the pseudo-regret when the bandit-reward (1) is defined over a single clique. The next theorem leverages the previous results to provide an upper bound on the joint pseudo-regret when the bandit-reward is defined over an arbitrary number of independent cliques in the DAG.

**Theorem 4** (Joint pseudo-regret in DAG-based games). *Consider a DAG-based game with bandit rewards given as in (1) and let $\mathcal{C}$ contain a collection of independent cliques associated with the DAG. Let each player operate according to Algorithm 1. Then, the joint pseudo-regret satisfies*

$$R(T) = \mathcal{O}\left(\sqrt{T \max_{k \in [|\mathcal{C}|]} |\mathcal{A}_{\mathcal{N}_k}|}\right)$$

*where $\mathcal{A}_{\mathcal{N}_k}$ denotes the joint action-space of the players in the kth clique $\mathcal{N}_k \in \mathcal{C}$.*

*Proof.* Let $\mathcal{N}_k \in \mathcal{C}$ denote the players belonging to the $k$th clique in $\mathcal{C}$ with joint action space $\mathcal{A}_{\mathcal{N}_k}$. The structure of (1) allows us to express the joint pseudo-regret as

$$R(T) = \mathbb{E}\left[\sum_{t=1}^{T} r_t(\mathbf{a}^\star) - r_t(\mathbf{a}(t))\right] \leq \sum_{k=1}^{|\mathcal{C}|} \beta_k \mathbb{E}\left[\sum_{t=1}^{T} r_t^k(\mathbf{a}_k^\star) - r_t^k(P^k(\mathbf{a}(t)))\right] \tag{10}$$

where

$$\mathbf{a}^\star = \arg\max_{\mathbf{a} \in \mathcal{A}_\mathcal{V}} \mathbb{E}\left[\sum_{t=1}^{T} r_t(\mathbf{a})\right], \quad \mathbf{a}_k^\star = \arg\max_{\mathbf{a} \in \mathcal{A}_{\mathcal{N}_k}} \mathbb{E}\left[\sum_{t=1}^{T} r_t^k(\mathbf{a})\right],$$

and the inequality follows since $\mathbb{E}\left[\sum_{t=1}^{T} r_t^k(P^k(\mathbf{a}^\star))\right] \leq \mathbb{E}\left[\sum_{t=1}^{T} r_t^k(\mathbf{a}_k^\star)\right]$. Now, for each clique $\mathcal{N}_k \in \mathcal{C}$, let the player indices in $\mathcal{N}_k$ be ordered according to the order of player observations within the clique. As Theorem 3 holds for any $\mathcal{N}_k \in \mathcal{C}$, we may, with a slight abuse of notation, bound the joint pseudo-regret of each clique as

$$R(T) \leq \sum_{k=1}^{|\mathcal{C}|} \beta_k R_{\mathrm{ub}}(T, \mathcal{N}_k) \leq \max_{k \in [|\mathcal{C}|]} R_{\mathrm{ub}}(T, \mathcal{N}_k)$$

where $R_{\mathrm{ub}}(T, \mathcal{N}_k)$ follows from Theorem 3 as

$$R_{\mathrm{ub}}(T, \mathcal{N}_k) = 4\sqrt{T} \sum_{i \in \mathcal{N}_k} \prod_{j \leq i, j \in \mathcal{N}_k} \sqrt{A_j} + \sum_{i \in \mathcal{N}_k^-} \prod_{j \leq i, j \in \mathcal{N}_k^-} A_j + 1$$

where $\mathcal{N}_k^-$ excludes the last element in $\mathcal{N}_k$. The result follows as $R_{\mathrm{ub}}(T, \mathcal{N}_k) = \mathcal{O}(\sqrt{T|\mathcal{A}_{\mathcal{N}_k}|})$. $\quad\square$

## 4   NUMERICAL RESULTS

The experimental setup in this section is inspired by the socio-economic simulation in (Zheng et al., 2020).[4] We consider a simple taxation game where one player acts as a socio-economic planner and the remaining $M$ players act as workers that earn an income by performing actions, e.g., constructing houses. The socio-economic planner divides the possible incomes into $N$ brackets where $[\beta_{i-1}, \beta_i]$ denotes the $i$th bracket with $\beta_0 = 0$ and $\beta_N = \infty$. In each round $t \in [T]$, the socio-economic planner picks an action $\mathbf{a}_p(t) = (a_{p,1}(t), \dots, a_{p,N}(t))$ that determines the taxation rate where $a_{p,i}(t) \in \mathcal{R}_i$ denotes the the marginal taxation rate in income bracket $i$ and $\mathcal{R}_i$ is a finite set. We use the discrete set $\mathcal{A}_p = \prod_{i=1}^N \mathcal{R}_i$ of size $A_p$ to denote the action space of the planner.

In each round, the workers observe the taxation policy $\mathbf{a}_p(t) \in \mathcal{A}_p$ and choose their actions consecutively, see Fig. 2a. Worker $j \in [M]$ takes actions $a_j(t) \in \mathcal{A}_j$ where $\mathcal{A}_j$ is a finite set. A chosen action $a_j(t) \in \mathcal{A}_j$ translates into a tuple $(x_j(t), \tilde{l}_j(t))$ consisting of a gross income and a marginal default labor cost, respectively. Furthermore, each worker has a skill level $s_j$ that serves as a divisor of the default labor, resulting in an effective marginal labor $l_j(t) = \tilde{l}_j(t)/s_j$. Hence, given a common action, high-skilled workers exhibit less labor than low-skilled workers. The gross income $x_j(t)$ of worker $j$ in round $t$ is taxed according to $\mathbf{a}_p(t)$ as

$$\xi(x_j(t)) = \sum_{i=1}^N a_{p,i}(t) \left[ (\beta_i - \beta_{i-1})\mathbf{1}\{x_j(t) > \beta_i\} + (x_j(t) - \beta_{i-1})\mathbf{1}\{x_j(t) \in [\beta_{i-1}, \beta_i]\} \right]$$

where $a_{p,i}(t)$ is the taxation rate of the $i$th income bracket and $\xi(x_j(t))$ denotes the collected tax. Hence, worker $j$'s cumulative net income $z_j(t)$ and cumulative labor $\ell_j(t)$ in round $t$ are given as

$$z_j(t) = \sum_{u=1}^t x_j(u) - \xi(x_j(u)), \quad \ell_j(t) = \sum_{u=1}^t l_j(u).$$

In round $t$, the utility of worker $j$ depends on the cumulative net income and the cumulative labor as

$$r_t^j(z_j(t), \ell_j(t)) = \frac{(z_j(t))^{1-\eta} - 1}{1 - \eta} - \ell_j(t) \tag{11}$$

where $\eta > 0$ determines the non-linear impact of income. An example of the utility function in (11) is shown in Fig. 2b for $\eta = 0.3$, income $x_j(t) = 10$, and a default marginal labor $\tilde{l}_j(t) = 1$ at different skill levels. It can be seen that the utility initially increases with income until a point at which the cumulative labor outweighs the benefits of income and the worker gets burnt out.

We consider bandit-rewards defined with respect to the worker utilities and the total collected tax as

$$r_t(\mathbf{a}_p(t), a_1(t), \dots, a_M(t)) = \frac{1}{(M+1)} \left[ \sum_{j=1}^M wr_t^j(z_j(t), \ell_j(t)) + w_p \sum_{j=1}^M \xi(x_j(t)) \right] \tag{12}$$

where the weights trade off worker utility for the collected tax and satisfy $Mw + w_p = M + 1$. The individual rewards are all normalized to $[0, 1]$, hence, $r_t(\mathbf{a}_p(t), a_1(t), \dots, a_M(t)) \in [0, 1]$.

For the numerical experiment, we consider $N = 2$ income brackets where the boundaries of the income brackets are $\{0, 14, \infty\}$ and the socio-economic planner chooses a marginal taxation rate from $\mathcal{R} = \{0.1, 0.3, 0.5\}$ in each income bracket, hence, $A_p = 9$. We consider $M = 3$ workers

---

[4]The source code of our experiments is available on `https://anonymous.4open.science/r/bandit_optimization_dag-242C/`.

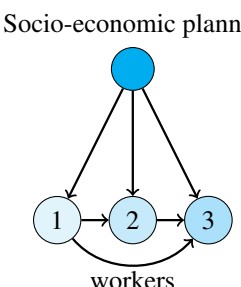

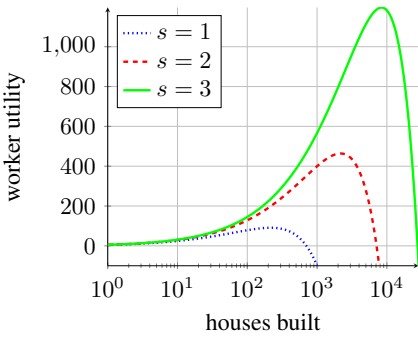

(a) Socio-economic setup with 4 players among which 3 are designated workers.

(b) Example of utility functions for different skill levels when $x_j(t) = 10$ and $\tilde{\ell}_j(t) = 1$.

Figure 2: Socio-economic setup.

with the same action set $\mathcal{A}$ of size 3. Consequently, the joint action space is of size 243. Furthermore, we let the skill level of the workers coincide with the worker index, i.e., $s_j = j$ for $j \in [M]$. Simply, workers able to observe others have higher skill. The worker actions translate to a gross marginal income and a marginal labor as $a_j(t) \rightarrow (x_j(t), l_j(t))$ where $x_j(t) = 5a_j(t)$ and $l_j(t) = a_j(t)/s_j$ for $a_j(t) \in \{1, 2, 3\}$. Finally, we set $\eta = 0.3$ and let $w = 1/M$ and $w_p = M$ to model a situation where the collected tax is preferred over workers' individual utility.

The joint pseudo-regret of the socio-economic simulation is illustrated in Fig. 3 along with the upper bound in Theorem 4. We collect 100 realizations of the experiment and, along with the pseudo-regret $R(T)$, two standard deviations are also presented. It can be seen that the players initially explore the action space and are able to eventually converge on an optimal strategy from a pseudo-regret perspective. The upper bound in Fig. 3 is admittedly loose and does not exhibit the same asymptotic decay as the simulation due to different constants in the scaling law, see Fig. 3b. However, it remains valuable as it provides an asymptotic no-regret guarantee for the learning algorithm.

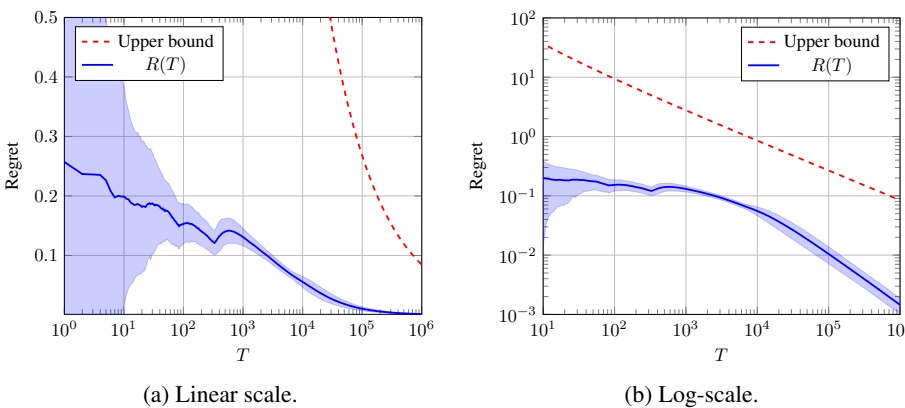

(a) Linear scale.

(b) Log-scale.

Figure 3: Pseudo regret vs the upper bound in Theorem 4.

## 5   CONCLUSION

We have studied multiplayer games with joint bandit-rewards where players execute actions consecutively and observe the actions of the preceding players. We introduced the notion of joint pseudo-regret and presented an algorithm that is guaranteed to achieve no-regret for both adversarial and stochastic bandit rewards. A bottleneck of many multi-agent algorithms is that the complexity scales with the joint action space (Jin et al., 2021) and our algorithm is no exception. An interesting venue of further study is to find algorithms that have more benign scaling properties, see e.g., (Jin et al., 2021; Daskalakis et al., 2021).

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
