# OpenReview forum: "Decentralized Online Bandit Optimization on Directed Graphs with Regret Bounds"
_ICLR.cc/2023/Conference — Submitted to ICLR 2023_

### Official Review · Reviewer_fbii · 2022-10-20

**Confidence:** 2
**Correctness:** 3
**Technical Novelty And Significance:** 2
**Empirical Novelty And Significance:** Not applicable
**Recommendation:** 3

**Clarity, Quality, Novelty And Reproducibility:**

The writing of this paper should be improved, as explained below.
1) The authors propose a new bandit setting, but do not clearly compare it against related existing bandit settings such as bandits with graph-structured feedback (Cesa-Bianchi et al., 2021). It is hard to justify what additional challenges the new setting brings.
2) In the Introduction of this paper, the concept of equilibrium has been mentioned many times. However, this paper only studies the regret bounds and does not study the equilibrium.
3) It seems that Step 8 of Algorithm 1 is implemented for all $j$, each of which will generate a strategy $\pi_i(t)$. It is not clear $\pi_i(t)$ generated by which $j$ is used by the player $i$ in Step 9 of Algorithm 1.
4) In Step 11 of Algorithm 1, the authors update the cumulative loss $L_{j,k}$. However, according to this update rule, it seems that $L_{j,k}$ for different $j$ have the same value. It is not clear why this is sufficient for running different bandit algorithms for all $j$.

Because of the writing problems in Algorithm 1, it is hard to strictly follow the proofs in this paper. For the experimental results, since the authors provide the source code, I believe it is reproducible.

Moreover, the novelty of this paper is limited because it seems that it is not difficult to extend the algorithm and theory of Zimmert & Seldin (2021) to the problem in this paper.

Overall, the quality of this paper may be below the bar of ICLR.


**Strength And Weaknesses:**

1) The proposed problem of decentralized online bandit optimization on directed graphs seems to be interesting.
2) The proposed algorithm is equipped with upper bounds on the joint pseudo-regret and its performance is partially verified by experimental results.

#Weaknesses
1) According to Algorithm 1, each player can observe the joint bandit reward, which may be unrealistic in the decentralized setting.
2) As in Eq. (1), the authors consider a joint bandit reward, in which each clique reward has a weight. However, according to Theorem 3, the upper bound of the proposed algorithm actually does not depend on the weight. There seem to be two possible explanations. First, weight is not important. Second, the proposed algorithm cannot reflect the importance of weight.
3) According to the analysis of this paper, it seems that it is not difficult to extend the algorithm and theory of Zimmert & Seldin (2021) into the problem in this paper, which limits the novelty of this paper.

**Summary Of The Paper:**

This paper formulates a problem of decentralized online bandit optimization on directed graphs, in which the graph structure dictates the order of the players and how players observe the actions of other players. Moreover, all players can observe a joint bandit reward, which is a linear combination of the reward of each clique in the graph. To solve this problem, the authors make an extension of an existing single-player multi-armed bandits algorithm and established upper bounds for the joint pseudo-regret.

**Summary Of The Review:**

The main contributions of this paper include a new bandit setting and a corresponding algorithm with pseudo-regret bounds. However, the writing of this paper should be improved, and the novelty of this paper seems to be limited. So, I tend to reject this paper.

---

### Official Review · Reviewer_8PKf · 2022-10-24

**Confidence:** 3
**Correctness:** 3
**Technical Novelty And Significance:** 2
**Empirical Novelty And Significance:** 2
**Recommendation:** 3

**Clarity, Quality, Novelty And Reproducibility:**

The writing should be improved as mentioned in the weaknesses. Both technical and empirical novelties are limited. Overall, the quality of this paper is a bit below the bar.

**Strength And Weaknesses:**

Strength:

The studied model is novel. The proposed algorithm achieves sublinear regrets.

Weaknesses:

1. The motivation of the studied model is not clear. Why do players have to execute actions consecutively? Why does the model consider the linear combination of the clique-rewards rather than all players' rewards? I suggest authors provide some real applications for the proposed model.
2. I'm confused about the definition of the independent clique in the DAG. I do not find the corresponding definition in (Koller & Friedman 2009, Ch.2). To increase the paper's readability, I suggest the authors provide the independent clique's definition.
3. I'm also confused about inequality (5). Notice that Theorem 1 requires the player to play T consecutive rounds. However, the rounds in the set $T_a$ are not consecutive. Thus (5) cannot be obtained by Theorem 1 directly. Am I right?
4. Tsallis-INF can achieve log(T) regret for stochastic rewards, but the proposed methods do not achieve log(T) regret in the stochastic setting.
5. The technical contribution is not strong enough. Most analyses are straightforward and the results are not surprising.
6. The authors only perform experiments on the graph with one clique. It would be better if the authors could provide numerical results on various connection graphs.

**Summary Of The Paper:**

This paper studies decentralized bandits on a DAG. In this model, the players perform actions consecutively and observe the actions of the preceding players. The paper proposes a novel algorithm for this model and achieves sublinear regret for both stochastic and adversarial rewards.


**Summary Of The Review:**

Though the studied model is novel, many details should be clarified (see weaknesses). Also, both theoretical and empirical contributions are a bit below the bar.

---

### Official Review · Reviewer_ibWi · 2022-10-25

**Confidence:** 4
**Correctness:** 4
**Technical Novelty And Significance:** 4
**Empirical Novelty And Significance:** 3
**Recommendation:** 8

**Clarity, Quality, Novelty And Reproducibility:**

Clarity: This paper is clear and easy to follow. It uses well denoted mathematics with good clarity.
Quality: This paper's results are trustworthy
Novelty: This paper studies a novel problem and proposes good solutions to the problem and the solutions are not that straightforward.
Reproducibility: I think the results (both theoretical and numerical) in this paper are reproducible.

**Strength And Weaknesses:**

Strengths: The graph-structure bandit problem studied in this paper is interesting and meaningful, and I am convinced that it is a missing part of the literature and can found potential applications in the real-world systems. This paper gives an algorithm with reasonable regret upper bounds, which I think is optimal or near optimal (ignoring constant factors).

Weakness: It is not sure whether the algorithm proposed in this paper is close to the lower bound. It will be much better if there can be analysis on the lower bound. Also, the numerical simulations are of small scales, but I think many real-world scenarios (such as communication networks) should have much larger scales. It will strengthen the paper a lot if there can be more large-scaled simulations.

Minor: Also wonder if the algorithm proposed in this paper can work only using local information. The step observing the reward seems to be costly in terms of communication in a large scale.

**Summary Of The Paper:**

This paper studies the regret bound of multi-player online bandit problems with a leader-follower (directed graph) hierarchical structure. To be specific, there are n players in total and a directed graph G, and for any two players with j having an out-edge to i, player i will observe the action of player j and take actions after player j. The reward is taken according to the joint actions taken all the players and can be either stochastic or adversarial.
With the above problem, the authors propose an algorithm, and proved the upper bound for single player and multi-players. For the single-player case, the regret upper bound is O(\sqrt{AT}) where A is the number of actions of this player and T is the time horizon, which I believe is optimal (up to a constant factor) as the regret bound's order is the same as the standard MAB problem. For multiple-player cases (n > 1), the regret is of order O(\sqrt{T\sum{A_i}}) where A_i is the action space (number of joint actions) of the i-th clique.
Finally the authors did some numerical simulations for the algorithm, and the results look consistent with the theory.

**Summary Of The Review:**

This paper studies an interesting and novel problem, and proposes an algorithm with good regret performance. The algorithm looks not trivial and the regret results are well supported by mathematical proofs. So I tend to vote an accept.

---

### Official Review · Reviewer_uQZP · 2022-10-25

**Confidence:** 3
**Clarity, Quality, Novelty And Reproducibility:** The paper is clearly written and the …
**Correctness:** 4
**Technical Novelty And Significance:** 4
**Empirical Novelty And Significance:** 2
**Recommendation:** 6

**Strength And Weaknesses:**

+ The paper considers a novel decentralized online bandit optimization setting, motivated by cooperative Stackelberg game.
+ The paper is well written. The results are novel and interesting.

-  The possible applications for the proposed setting is a bit vague.
-  Although a few theorems are presented, Theorem 4 is the most general one which covers the previous ones.


**Summary Of The Paper:**

The paper considers a decentralized multiplayer game with a leader-follower hierarchy described by a directed acyclic graph. It proposes a cooperative learning algorithm and proves that the algorithm achieves sub-linear joint pseudo-regret for both adversarial and stochastic bandit rewards.

**Summary Of The Review:**

The paper considers a decentralized multiplayer game with a leader-follower hierarchy described by a directed acyclic graph. It proposes a cooperative learning algorithm and proves that the algorithm achieves sub-linear joint pseudo-regret for both adversarial and stochastic bandit rewards. The paper is well written. The results are novel and interesting. The possible applications for the proposed setting is a bit vague.

---

### Decision · Program_Chairs · 2023-01-20

**Decision:**

Reject

**Justification For Why Not Higher Score:**

The main reason is limited novelty.

**Justification For Why Not Lower Score:**

N/A

**Metareview: Summary, Strengths And Weaknesses:**

This paper studies a bandit problem with a hierarchy of players dictated by a directed graph. The problem is studied in both stochastic and adversarial settings, and the authors derive sublinear regret bounds for their algorithms. The algorithms are evaluated on simple synthetic problems. The rebuttal was not submitted. This is likely because two reviews suggest a rejection. The main reason is limited novelty, given a plethora of works on bandits and graphs.